# Study on the Effects of Printing Process and Reinforcement Materials on the Performance of 3D-Printed Glass Bead Insulation Mortar

**DOI:** 10.3390/ma17133220

**Published:** 2024-07-01

**Authors:** Jun Wang, Xuanzheng Zhang, Zhenhua Liu, Jiasheng Zhao

**Affiliations:** College of Civil Engineering, Henan University of Technology, Zhengzhou 450001, China; zhangxuanzheng2023@163.com (X.Z.); 13253372294@163.com (Z.L.); jiasheng_zhao2023@163.com (J.Z.)

**Keywords:** 3D printing, printing process, reinforcement materials, insulation mortar, performance

## Abstract

Based on 3D printing technology, this paper investigates the effects of the printing process and reinforcement materials on the performance of 3D-printed glass bead insulation mortar. In order to improve and enhance the performance of the mortar, two sets of tests were designed for research and analysis. Firstly, by changing the direction of the interlayer printing strips, the anisotropy of the specimens in different paths was analyzed, and then the effect of different dosages of different fibers on the performance of 3D-printed glass bead insulation mortar was investigated by adding reinforcing materials. The results show that the path a specimen in the X direction’s compressive strength is the best; in the Y direction, flexural strength is the best; the path b specimen in the Y direction’s compressive strength is the best; in the Z direction, flexural strength is the best, but the compressive and flexural strengths are lower than the strength of the specimen without 3D printing (cast-in-place specimen); and adding reinforcing materials mortar not only has high strength but also has good printability and excellent thermal insulation. This paper provides a theoretical basis and reference value for the popularization and application of 3D printing thermal insulation mortar technology.

## 1. Introduction

With the rapid development of the information age, the construction industry is also gradually moving towards intelligence, and the concrete 3D printing construction technology, as a typical intelligent construction process, which has the advantages of being modeless, flexible, fast, and green, providing environmental protection, and having high efficiency and economy, has been favored by the construction industry at home and abroad.

Different printing processes will lead to different anisotropy, and many scholars have set the printing parameters as a fixed value and conducted a series of studies on 3D printing anisotropy. For instance, Biranchi Panda et al. [1] investigated the anisotropic mechanical properties of 3D printing, and the experimental results showed that the compressive strength in the X direction was the strongest, and the flexural strength in the Z direction was the strongest. Ma et al. [2] concluded that for the suitable fiber doping amount based on the appropriate fiber admixture, the compressive strength was measured to be the highest in the X direction, but the flexural strength was weaker, and the flexural strength was the strongest in the Z direction, but the compressive strength was inferior. Xiao et al. [3] carried out a finite element analysis on the compressive and flexural anisotropy of 3D-printed concrete, which is consistent with the conclusions drawn by the above mentioned scholars, and similarly yielded the highest compressive strength in the X direction. The flexural strength was relatively weaker, and the compressive strength of specimens loaded in the Y and Z directions was comparatively low, whereas the flexural strength was much higher.

In addition to the printing process, the inclusion of reinforcing materials has a significant improvement on the performance of concrete, and due to the specificity of 3D printing technology, it is difficult to realize the synchronous placement of reinforcement in the printing process. Therefore, solving the problem of the integrated printing of steel reinforcement in the construction process is a difficult point to achieve the development of automation and intelligence in the field of construction, and some scholars have enhanced and toughened concrete materials by adding various fibers and controlling the directional distribution of fibers [4,5,6]. According to statistics, residential energy consumption accounts for a large part of global energy consumption, and with the improvement of the living standards of residents, building consumption will further increase [7]. Studies have shown that the addition of glass beads to concrete materials can effectively improve thermal insulation to reduce indoor and outdoor heat exchange and thus achieve the purpose of energy saving [8]. As the glass beads are very easy to break during the mixing process due to extrusion or subjected to vibration, it is easy to cause strength loss [9]. Relevant scholars through research found that the addition of fibers and glass beads after the mixing of concrete can improve the strength as well as durability, and after the destruction of the phenomenon, it can be formed cracked but not dispersed [10]. Based on the above research background, this test selected polypropylene fiber and basalt fiber as reinforcing materials, polypropylene fiber has the advantages of a low self-weight, good crack resistance, and low price, and it has the characteristics of high fineness and quantity, and so on. It plays a toughening role in order to improve the mechanical properties of mortar [11], and basalt fiber has a small coefficient of thermal conductivity, high modulus of elasticity, tensile strength, dispersion, compatibility with cementitious materials, and the ability to be used as a reinforcing material. Basalt fiber has the advantages of low thermal conductivity, high elastic modulus, high tensile strength, good dispersion, good compatibility with cementitious materials, low cost, wide sources, etc. It has attracted wide attention at home and abroad [12].

In summary, this paper compared previous scholars’ methods in order to obtain a mortar with high strength and good thermal insulation performance, based on 3D printing technology using glass beads thermal insulation mortar as the object for research. First of all, we determined the material ratio, with cast-in-place concrete as a comparison, in order to evaluate the anisotropy of the printing path a, b. After the glass bead insulation mortar strength due to the defective nature of the glass bead insulation mortar, we prepared four kinds of polypropylene fiber with different mixing amounts (0.3%, 0.6%, 0.9%, 1.2%) and basalt fiber mortar for the analysis of the performance of the comparison. This study concluded that the mortar had good printability, excellent heat preservation performance, and high strength. The test results of this paper provide a theoretical basis and reference value for the popularization and application of 3D printing thermal insulation mortar technology.

## 2. Materials and Methods

### 2.1. Material Apparatus and Test Ratio

#### 2.1.1. Material

Since this paper investigates the influence of printing process and reinforcement materials on the performance of 3D-printed thermal insulation mortar, i.e., two groups of tests, the former is named Test 1 and the latter is named Test 2 for ease of differentiation (the same below).

Test 1 used P.O42.5 common silicate cement produced by Zhucheng Yangchun Cement Co. (Weifang, China). The chemical composition is given in Table 1; Fly ash: Grade I fly ash produced by Henan Hengyuan New Material Co. (Xinyang, China); Hydrated lime: Analytically pure calcium hydroxide produced by Tianjin Zhonglian Chemical Reagent Co. (Tianjin, China); Aggregate: glass beads produced by Haohui Thermal Insulation Material Factory (Langfang, China); Hydroxypropyl methylcellulose ether: manufactured by Shanghai Kingland Source Chemical Technology Co. (Shanghai, China); Gum Powder: Hongsheng Environmental Protection New Material Co., (Huaian, China).

Test 2 adds polypropylene fibers and basalt fibers produced by Jiangsu Tianyi Engineering Fiber Co. (Changzhou, China). on the basis of the material of Test 1, performance as a reinforcing material for this test is shown in Table 2 and Table 3, and their appearance can be seen in Figure 1.

#### 2.1.2. Testing Equipment

Test 1 and Test 2 were conducted using HC-3DPRT/LS concrete (mortar) 3D printer produced by Hangzhou Guanli Intelligent Technology Co., Ltd. (Hangzhou, China), NLD-2 cementitious sand flow tester produced by Wuxi Jianyi Instrument Machinery Co. (Wuxi, China). Thermal Conductivity Instrument, JYE-2000A Automatic Constant Stress Pressure Tester produced by Beijing Kedar Jingwei Science and Technology Development Co., Ltd. (Beijing, China), DKZ-5000 Electric Flexural Tester produced by Tianjin Gangyuan Experimental Instrument Factory (Tianjin, China), and 101 Electric Heating Blast Drying Oven, Constant Temperature and Humidity Curing Room, Concrete Trial Mold, Electronic Scale, Vibrating Rod, and Scraper produced by Beijing Yongmingming Medical Instrument Factory (Beijing, China).

### 2.2. Specifications

Test 1 prepared the printing performance of excellent glass beads’ insulation mortar fit ratio, as shown in Table 4. Through the printable glass beads insulation mortar compression test, flexural test, and thermal conductivity test, we analyzed the anisotropy on the mechanical properties and thermal insulation performance of 3D printing glass beads insulation mortar.

Test 2 is based on Table 4 to add different fibers in different doses to increase the strength of glass bead insulation mortar to adapt to different environments and requirements. We add fiber as reinforcing material to obtain different fibers and different dosages of glass bead insulation mortar flow, printability, dry density, flexural strength, compressive strength, and thermal conductivity of the influence of the law. This study concluded with good printability, excellent thermal insulation, and high strength of the insulation mortar. The added fiber length was 6 mm and dosages were 0.3%, 0.6%, 0.9%, and 1.2%, with K0 indicating the benchmark group, and PP1, PP2, PP3, PP4, BF1, BF2, BF3, BF4, respectively, indicating the polypropylene fiber and basalt fiber dosages. The experimental design of the ratio is shown in Table 5.

### 2.3. Setting of Printer Parameters

The molding quality and molding accuracy of 3D-printed components are not only related to the printing equipment and printing materials but also affected by the printer parameters. Under the premise of certain printing equipment and printing materials, different printing parameters will cause different printing effects [13]. Among them, nozzle diameter, printing speed, extrusion speed, and printing path are important factors that affect the molding quality and molding accuracy of 3D-printed components [14].

Through a large number of 3D printing test piece explorations, the printing parameters are finally set to a nozzle diameter of 20 mm, a print layer height of 10 mm, a print speed of 0.7, and an extrusion speed of 1. Test 1 adopts two print paths: print path a, which is parallel to the print strips between the layers, and print path b, which is perpendicular to the print strips between the layers, as shown in Figure 2.

### 2.4. Test Block Design

Test 1: The concrete 3D printer can be divided into three directions according to the printing process, in which the X direction is the traveling direction of the print head, the Y direction is the direction between strips, and the Z direction is the direction between layers. Referring to the China Building Materials Association standard T/CCPA 33-2022 “Test Methods for Basic Mechanical Properties of 3D Printed Concrete” [15] and GB/T 5486-2008 “Test Methods for Inorganic Rigid Adiabatic Products” [16], the standard test blocks for this test are 70.7 × 70.7 × 70.7 mm standard cubic test blocks and 40 × 40 × 160 mm standard rectangular specimen blocks, which were put into a standard curing room with a temperature of (20 ± 2) °C and a relative humidity of more than 95% for curing. Among them, the cast-in-place specimen blocks were cast using standard molds; due to the large influence of the concrete 3D printer by the printing parameters, the edges of the printed components have a certain curvature, in order to minimize the influence of other factors, all the 3D-printed specimen blocks of the test were sampled from a single 3D-printed rectangle, and the rectangle of 250 × 450 × 90 mm was prepared for compressive strength specimen sampling. Rectangles of 280 × 320 × 60 mm were used for sampling of flexural strength specimens.

Considering the dimensions of different printing paths and different printing directions, the effects of anisotropy on the mechanical properties and thermal conductivity of 3D-printed components were investigated by compressive test, flexural test, and thermal conductivity test, and the specific test groups are shown in the following Table 6, Table 7 and Table 8.

Test 2: The standard test specimen for this test is a standard cubic specimen of 70.7 × 70.7 × 70.7 mm and a standard rectangular specimen of 40 × 40 × 160 mm, which is put into a standard curing room with a temperature of (20 ± 2) °C and a relative humidity of more than 95% for curing.

### 2.5. Test Block Fabrication and Testing

Test 1: The 3D-printed specimens are cut and sampled by printing 250 × 450 × 90 mm and 280 × 320 × 60 mm rectangles on site. Before cutting, the print specimen should be placed in the standard maintenance room for at least seven days. Cutting should be removed from the print end and truncation of the end face so that the interface position is located as far in the middle of the specimen as possible or so that the print layer and the print strips in the specimen are symmetrically distributed. Cutting should be timely in the specimen surface marking X, Y, and Z. Cutting is completed after the test specimen continues to be placed into the standard maintenance room for maintenance until the end of the test period. The specific situation is shown in Figure 3.

Cast-in-place specimen adopts standard test mold for casting; before casting, the inside of the test mold should be coated with lubricant, and the mold should be dismantled within 48 h after casting and then put into the standard curing room for curing until the end of the test period, as shown in Figure 4.

The test method of cast-in-place specimen compressive, flexural, and thermal conductivity refers to the previous Section 2.3. 3D-printed specimen has anisotropy, which can be divided into X, Y, and Z directions according to the printing direction, in which the loading method of compression resistance is shown in Figure 5.

Test 2: This test tested the different reinforcing materials on mortar fluidity and dry density before curing, as well as the determination of compressive strength, flexural strength, and thermal conductivity after curing was completed, and the determination method was the same as that of Test 1.

## 3. Results and Discussion

### 3.1. Test 1 Results and Analysis

#### 3.1.1. Compressive Strength Anisotropy Analysis

The cut 3D-printed specimen block is subjected to a compressive test to obtain the compressive strength in the X, Y, and Z directions of different printing paths, as shown in Figure 6.

An analysis of Figure 6 shows that the compressive strength values of the printed specimens in the X, Y, and Z directions under different printing paths are lower than those of the cast-in-place specimens, which may be due to the fact that the 3D-printed specimens are made of layers of printed strips stacked on top of each other, which causes a large number of pores in the process of stacking, and the formation of interlayer weak surfaces and interstrip weak surfaces, which negatively affect the mechanical properties of the specimens. The compressive strength of the specimen printed in path a in the X direction is greater than that in the Y and Z directions, which may be due to the fact that when the specimen is loaded in the X direction, the loading direction is the same as the tangent direction of the interlayer interface and the interstrip interface inside the specimen, and the printed strip will not undergo relative slip damage under the loading load, and each printed strip can give full play to its own load carrying capacity, so the compressive strength is higher. In contrast, when loading in the Y and Z directions, the loading direction is perpendicular to the direction of the weak surface, which may cause relative slip between the printed strips and affect the compressive strength of the specimen. Compared with path a, path b improved the compressive strength of the printed specimen in the X, Y, and Z directions to a certain extent, which may be attributed to the fact that the interlayer vertical printing reduces the gaps and weak surfaces of the specimen, which makes the inside of the printed specimen denser. At the same time, the interlayer cross vertically arranged print strips can effectively reduce the formation of through-seams and better control the development of cracks and the occurrence of lateral slip damage. Path b specimen has the largest compressive strength in the Y direction and the smallest compressive strength in the Z direction. A comparative analysis of the compressive strengths of path a and path b in the X, Y, and Z directions, respectively, reveals that the change in compressive strength of path a print specimen in the X, Y, and Z directions is greater than that of path b, which indicates that path b has a certain attenuating effect on the anisotropy of the print specimen.

#### 3.1.2. Flexural Strength Anisotropy Analysis

The cut 3D-printed specimen block is subjected to flexural test to obtain the flexural strength in X, Y, Z directions of different printing paths, as shown in Figure 7.

Analysis of Figure 7 shows that the flexural strength of 3D-printed specimens is lower than that of cast-in-place specimens in different loading directions, which may be due to the existence of a large number of gaps and weak surfaces inside the 3D-printed specimens, which weaken the flexural strength of the specimens. The lowest flexural strength of path a printed specimen in the X direction may be due to the existence of weak interfaces perpendicular to the tensile stresses in the specimen in the X direction, which ultimately leads to a reduction in the flexural strength of the specimen. There is not much difference in the flexural strength in the Y and Z directions, which may be due to the fact that the interfaces of the Y- and Z-direction specimens have a relatively similar distribution. However, the flexural strength of the Y-direction specimen is slightly larger than that of the Z-direction specimen, which may be due to the fact that the interface perpendicular to the loading surface in the Y-direction specimen is inter-bar interface, while the interface perpendicular to the loading surface in the Z-direction specimen is inter-layer interface, which is more densely distributed than the inter-bar interface, and so the final result of Y-direction specimen has slightly larger flexural strength than that of Z-direction specimen. The average flexural strength of the printed specimen of path b is slightly higher than that of the printed specimen of path a. This may be due to the fact that the layers of the printed strips of path b are perpendicular to each other, so that the loading direction of the specimen is not parallel to the normals of some of the interfaces, which weakened the influence of the weak interfaces between the layers, and thus the flexural strength of the printed specimen of path b is increased. Comparison of the flexural strength of the path b specimens from the X, Y, and Z directions reveals that the best flexural strength of the specimens is in the Z direction, followed by the Y and X directions.

#### 3.1.3. Thermal Conductivity Anisotropy Analysis

The cut 3D-printed test blocks are tested for thermal conductivity to obtain the thermal conductivity of different printing paths in X, Y, and Z directions, as shown in Figure 8.

An analysis of Figure 8 shows that the thermal conductivity of the printed specimen is lower than that of the cast-in-place specimen, which may be due to the fact that the printed specimen is piled up by the strips, which will form a large number of gaps and interlayer weak surfaces, and there is no vibration process in the printed specimen, and it is difficult for the air bubbles to be discharged efficiently, which leads to a larger porosity inside the specimen, so the thermal conductivity of the printed specimen is lower than that of the cast-in-place specimen. The average value of the thermal conductivity of the test piece printed in path b is smaller than the average value of the thermal conductivity of the test piece printed in path a. This may be due to the fact that the interlayer printing strips in path b are perpendicular to each other, and the printing strips will form a mutual compensatory filling effect, which reduces defects brought about by 3D printing, and reduces the internal porosity of the test piece, resulting in the test piece with a large thermal conductivity. The change in the thermal conductivity of the test piece in the X, Y, and Z directions of the 3D-printed test piece is not significant. The thermal conductivity of the 3D-printed specimen in X, Y, and Z directions does not change much, which may be due to the fact that the thermal conductivity is mainly related to the porosity inside the specimen when the material is unchanged, and the larger the porosity is, the smaller the thermal conductivity is. The porosity inside the test piece will not change after printing, so the thermal conductivity of the 3D-printed test piece does not change significantly with the change of test direction.

### 3.2. Test 2 Results and Analysis

#### 3.2.1. Mortar Flow Test Results and Analysis

The effect of different polypropylene fiber dosages and basalt fiber dosages on the flow of 3D-printed glass beads insulation mortar is shown in Figure 9.

An analysis of Figure 9 shows that with the increase of fiber doping, the mortar fluidity gradually decreases, which may be due to the fact that the fiber is dispersed in the mortar, which will form a three-dimensional reticulation structure, forming a fixed effect on glass beads, water, and other liquid substances, with a certain degree of water retention, which leads to a decrease in the fluidity of the mortar. From the figure, it can also be seen that the flow degree of basalt fiber is lower than that of polypropylene fiber under the same fiber dosage. This may be because polypropylene fiber is flexible. Its shape can be changed arbitrarily with the mixing process, so the material is relatively dense, while the basalt fiber is a rigid material. In the mixing process, it needs to consume part of the mortar to wrap it, which leads to a reduction in the flow of material in the mortar, so that the flow of mortar is lower than that of polypropylene fiber under the same dosage.

#### 3.2.2. Dry Density Test Results and Analysis of Mortar

The effect of different polypropylene fiber dosages and basalt fiber dosages on the dry density of 3D-printed glass bead insulation mortar is shown in Figure 10.

Analysis of Figure 10 shows that the effect of fiber on the dry density of mortar is small. Overall, the dry density of mortar increases with the increase of fiber dosage, which may be due to the fact that when the fiber dosage is less, it has a certain filling effect on the voids in the mortar, making the mortar denser, which leads to an increase in the dry density of the mortar. In addition, in the same fiber dosage, adding polypropylene fiber mortar dry density is less than adding basalt fiber mortar dry density, which may be due to the basalt fiber itself being denser, so the addition of basalt fiber mortar density is greater than the addition of polypropylene fiber mortar density.

#### 3.2.3. Mortar Compressive Strength Test Results and Analysis

The effect of different polypropylene fiber dosages and basalt fiber dosages on the compressive strength of 3D-printed glass bead insulation mortar is shown in Figure 11.

An analysis of Figure 11 shows that the compressive strength of the mortar increases and then decreases with the increase in polypropylene fiber dosage, and the compressive strength of the mortar reaches a maximum of 1.49 MPa when the dosage is 0.6%. The increase in compressive strength of the mortar may be due to the fact that the fibers are three-dimensional mesh-like structures inside the mortar, which can play a very good role in connecting and supporting the development of cracks with an inhibitory effect and thus improve the compressive strength of the mortar. Compressive strength of mortar. In the polypropylene fiber dosage of more than 0.6%, the compressive strength of mortar with the increase in fiber dosage is a decreasing trend, which may be due to the fiber dosage being too much, the mortar mixing not being uniform, the fiber being easy to agglomerate inside the mortar, and the formation of weak points, leading to the compressive strength of the mortar being reduced. The compressive strength of the mortar increases with the increase of basalt fiber dosage, and the compressive strength of the mortar reaches a maximum of 1.62 MPa when the dosage is 1.2%, due to the better dispersion of basalt fibers compared to polypropylene fibers, which are relatively less likely to agglomerate in the mortar, so the compressive strength curve does not appear inflection point.

#### 3.2.4. Flexural Strength Test Results and Analysis of Mortar

The effect of different polypropylene fiber dosages and basalt fiber dosages on the flexural strength of 3D-printed glass bead insulation mortar is shown in Figure 12.

Figure 12 shows that the change rules for different fiber dosages’ impact on mortar flexural strength and mortar compressive strength change rules are basically the same. With the increase of polypropylene fiber doping, the mortar flexural strength first increased and then decreased; for the doping amount of 0.6%, the mortar flexural strength reached a maximum of 0.84 MPa. With the increase in basalt fiber doping, the mortar flexural strength gradually increased and an inflection point did not appear. For the doping amount of 1.2%, the mortar flexural strength reached a maximum of 0.89 MPa.

#### 3.2.5. Test Results and Analysis of Thermal Conductivity of Mortar

The effect of different polypropylene fiber dosages and basalt fiber dosages on the thermal conductivity of 3D-printed glass bead insulation mortar is shown in Figure 13.

An analysis of Figure 13 shows that polypropylene fiber has a negative effect on mortar thermal insulation performance. Mortar thermal conductivity increases with the increase of polypropylene fiber doping. When the polypropylene fiber doping is 0.6%, the mortar thermal conductivity rises by 8.5%. In addition, polypropylene fiber in the mortar is not easy to disperse, is susceptible to agglomeration phenomenon, and will form thermal bridges in the heat conduction process, reducing the thermal insulation performance of thermal insulation mortar. The doping of basalt fibers does not have a significant effect on the thermal conductivity of the mortar, which may be due to the low thermal conductivity of basalt fibers and their relatively easy dispersion in the mortar.

## 4. Conclusions

The main research results and conclusions of this paper are as follows:

Path a specimen has the best compressive strength in the X direction and the best flexural strength in the Y direction. Path b has the best compressive strength in the Y-direction and the best flexural strength in the Z-direction. The performance of 3D-printed specimens is also related to the print path, and different print paths will have different effects on the performance of specimens. The compressive and flexural strengths of path a (parallel printing strips between layers) are smaller than that of path b (vertical printing of interlayer strips), but the thermal conductivity of path a is larger than that of path b. The thermal conductivity of path a is larger than that of path b, and the thermal conductivity of path b is larger than that of path b.

As polypropylene fiber doping increases, insulation mortar fluidity, dry density, and thermal conductivity increase; mortar compressive strength and flexural strength with the increase in polypropylene fiber doping initially showed growth followed by a decline but the overall is still higher than the benchmark group. For the fiber doping of 0.6%, the compressive strength and flexural strength of the insulation mortar reached maximums, respectively, of 1.49 MPa and 0.84 MPa. With the increase in basalt fiber doping, the thermal conductivity of the mortar does not change much, though the fluidity and dry density tend to increase. The compressive and flexural strength of the mortar increases with the increase in basalt fiber doping, and the compressive and flexural strength of the mortar reach maximums of 1.62 MPa and 0.89 MPa, respectively, when the fiber doping is 1.2%.

## Figures and Tables

**Figure 1 materials-17-03220-f001:**
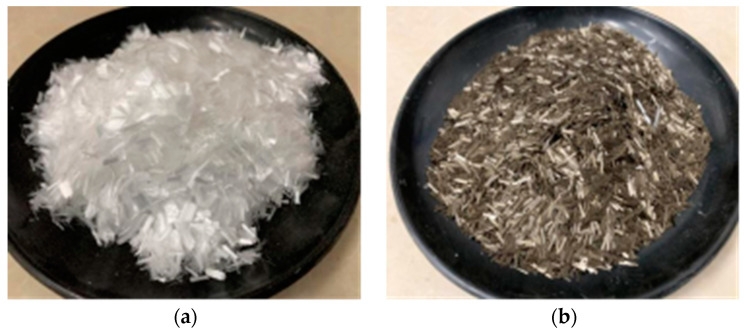
(**a**) Polypropylene Fiber; (**b**) Basalt Fiber.

**Figure 2 materials-17-03220-f002:**
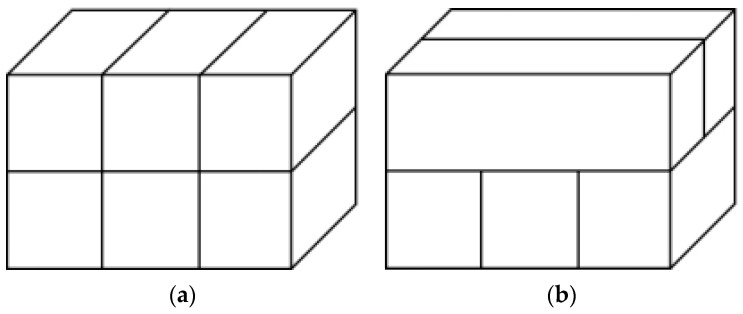
(**a**) Interlayer Print Strips Parallel; (**b**) Interlayer Print Strips Perpendicular.

**Figure 3 materials-17-03220-f003:**
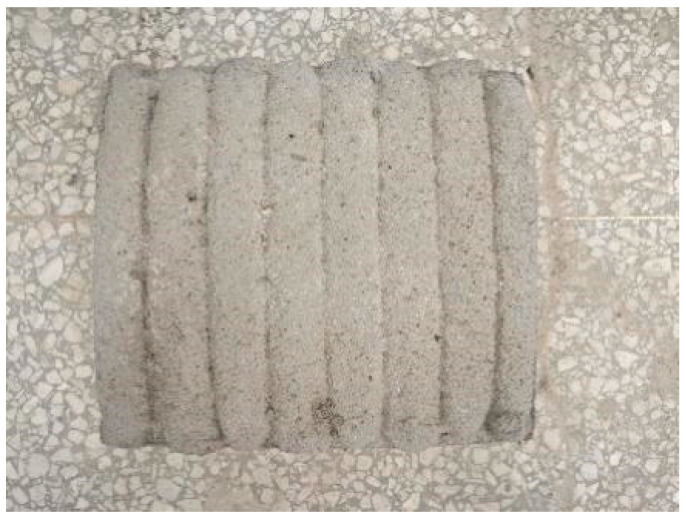
3D-printed test block.

**Figure 4 materials-17-03220-f004:**
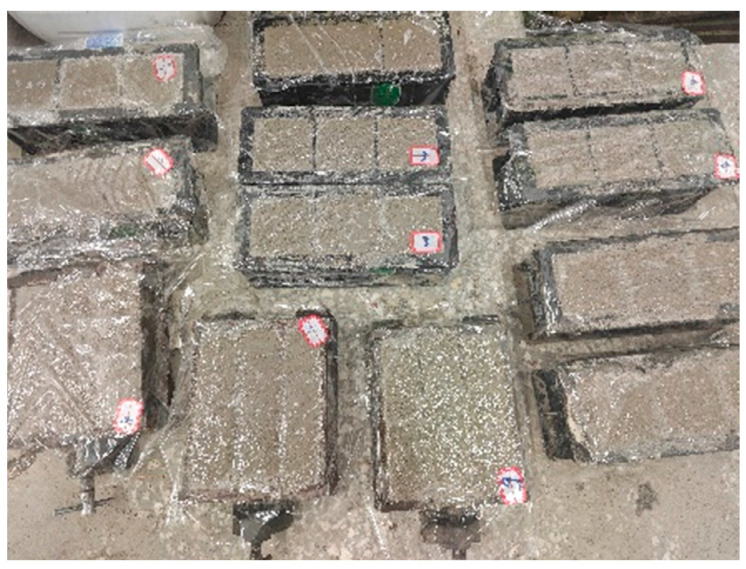
Cast-in-place test block.

**Figure 5 materials-17-03220-f005:**
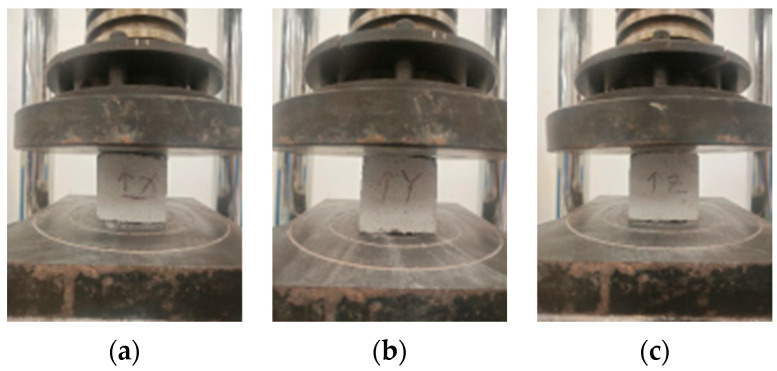
(**a**) X-Direction; (**b**) Y-Direction; (**c**) Y-Direction.

**Figure 6 materials-17-03220-f006:**
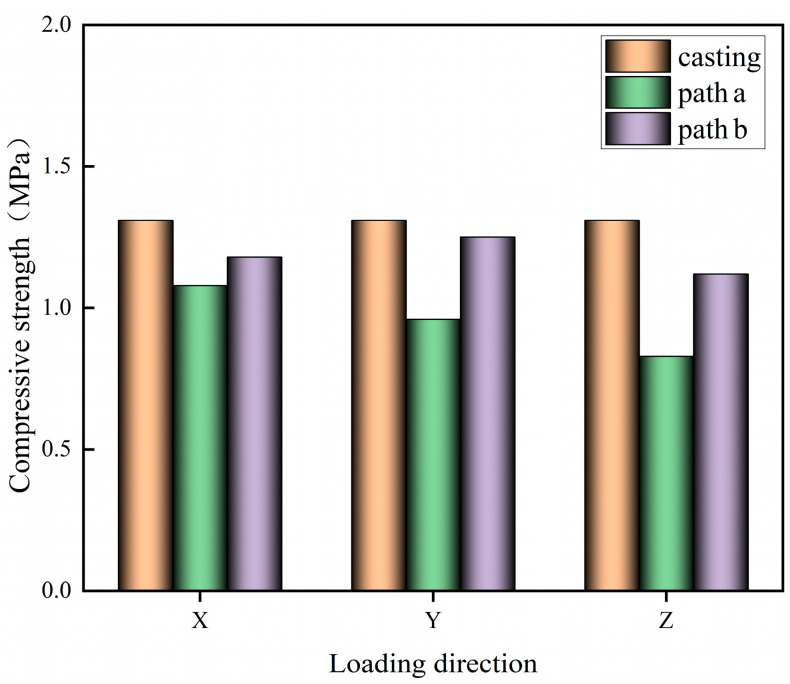
Compressive strength of specimen with different loading directions.

**Figure 7 materials-17-03220-f007:**
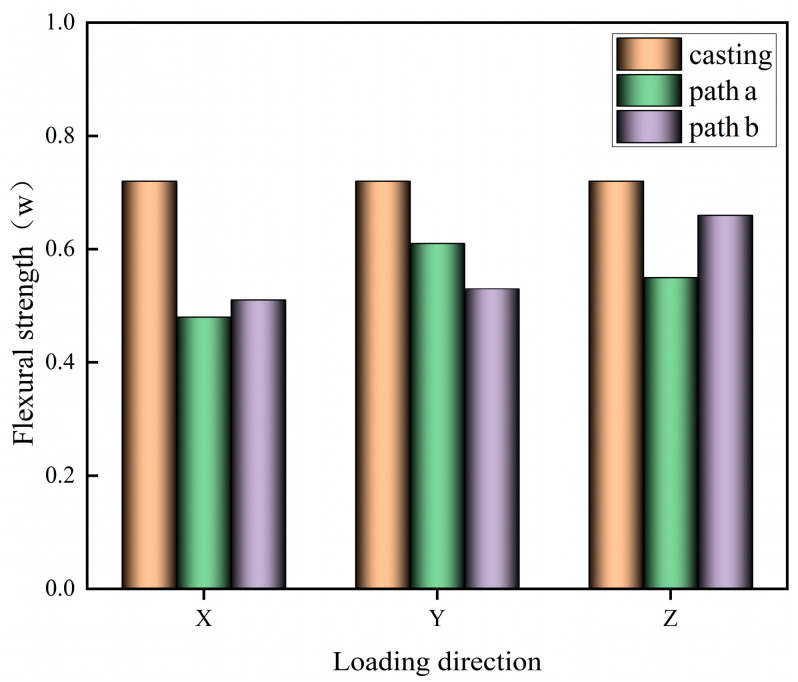
Flexural strength of specimen in different loading directions.

**Figure 8 materials-17-03220-f008:**
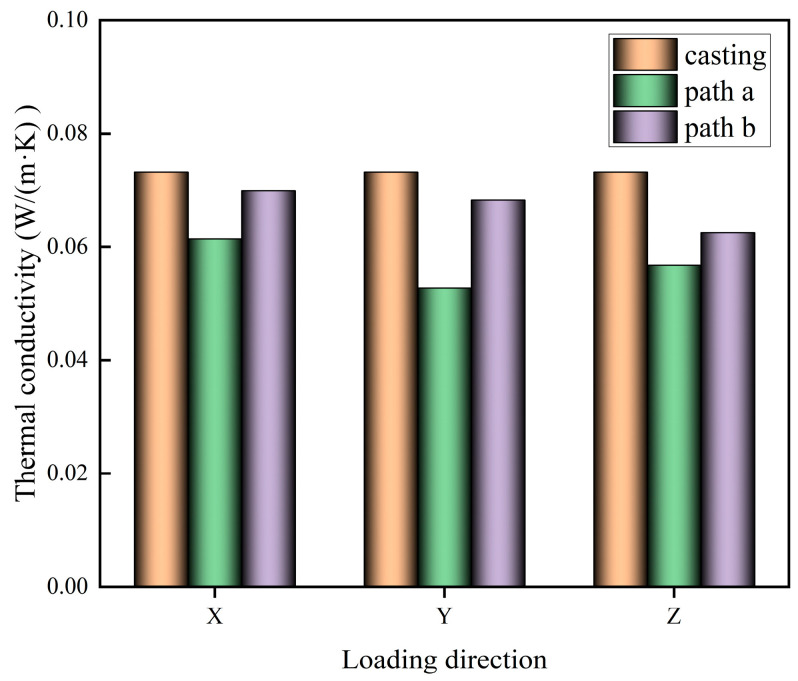
Thermal conductivity in different test directions.

**Figure 9 materials-17-03220-f009:**
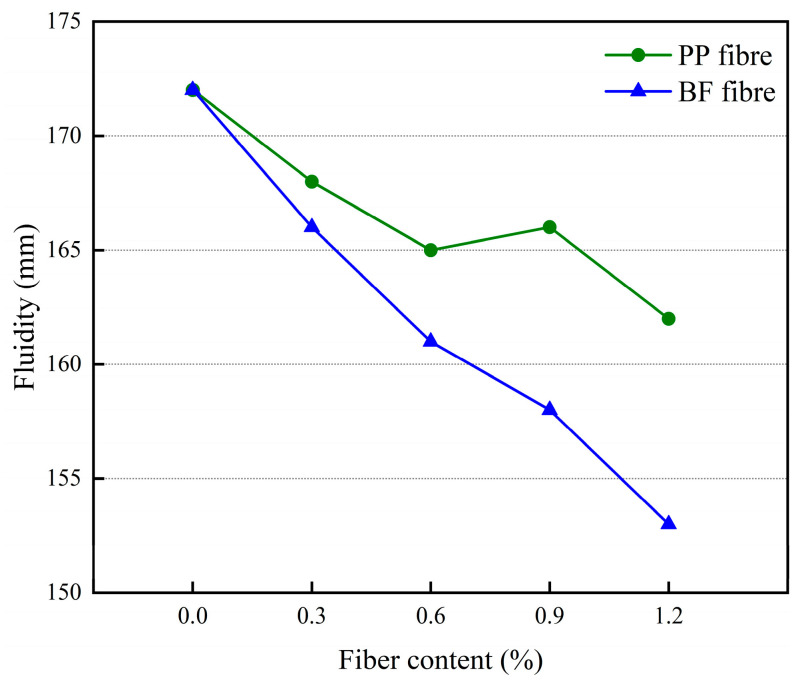
The effect of different fiber dosages on mortar fluidity.

**Figure 10 materials-17-03220-f010:**
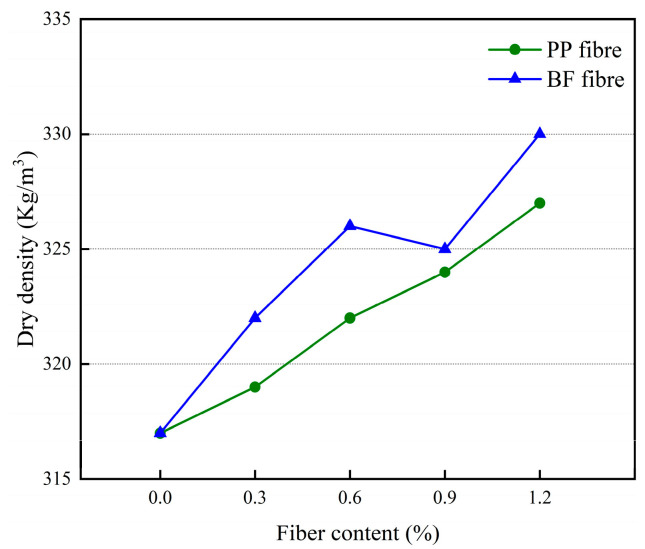
The effect of different fiber dosages on dry density of mortar.

**Figure 11 materials-17-03220-f011:**
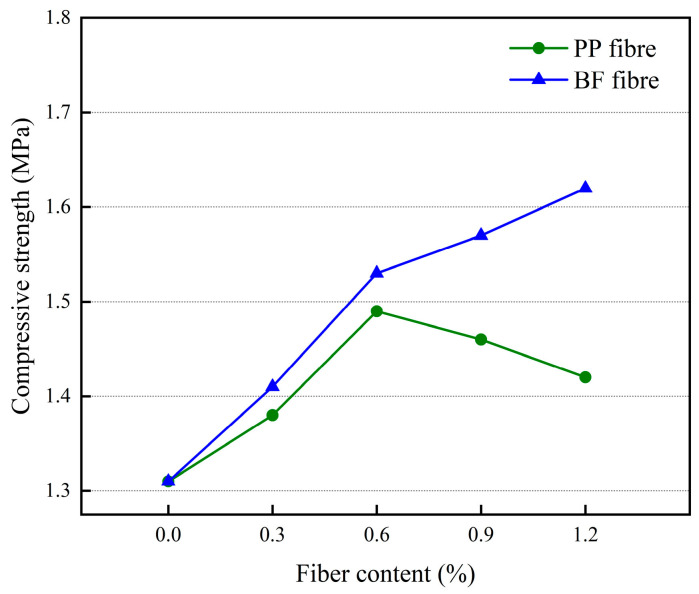
The effect of different fiber dosages on the compressive strength of mortar.

**Figure 12 materials-17-03220-f012:**
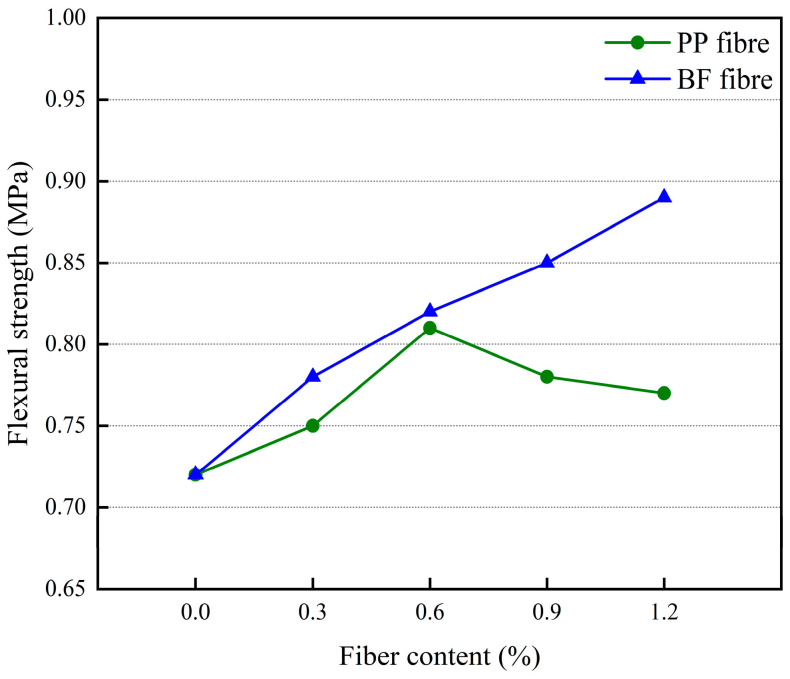
The effect of different fiber dosages on the flexural strength of mortar.

**Figure 13 materials-17-03220-f013:**
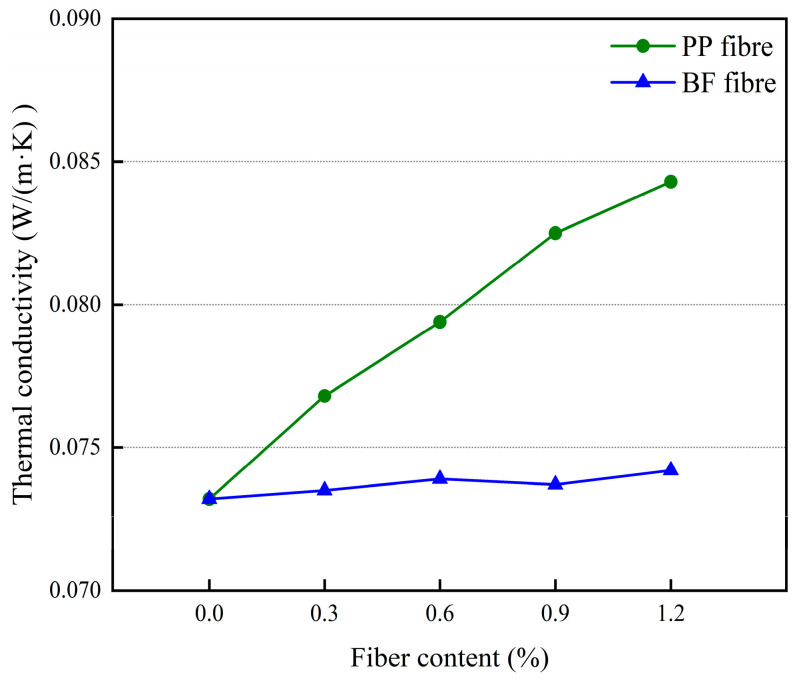
The effect of different fiber dosages on the thermal conductivity of mortar.

**Table 1 materials-17-03220-t001:** Common silicate cement chemical composition table.

Chemical Composition	SiO_2_	Al_2_O_3_	Fe_2_O_3_	CaO	MgO	SO_3_	Heat Loss
Content (%)	24.99	8.26	4.03	51.42	3.71	2.51	3.31

**Table 2 materials-17-03220-t002:** Performance parameters of polypropylene fiber.

Item	Length (mm)	Diameter (μm)	Density (g/cm^3^)	Tensile Strength (MPa)	Modulus of Elasticity (MPa)
performances	6	35	0.91	630	6000

**Table 3 materials-17-03220-t003:** Basalt fiber performance parameters.

Item	Length (mm)	Diameter (μm)	Density (g/cm^3^)	Tensile Strength (MPa)	Modulus of Elasticity (MPa)
performances	6	17	2.5	2300	86,000

**Table 4 materials-17-03220-t004:** Test mix ratio.

	Cementitious Material (g)			Glass Beads (g)	Gum Powder (g)	Cellulose Ether (g)	W/B
	Cement	Coal Ash	Slaked Lime				
1	750	200	50	1111.1	20	5	2.7

**Table 5 materials-17-03220-t005:** Different fiber dosage mortar mixing ratios.

	Cementitious Material (kg)			Glass Beads (kg)	Gum Powder (kg)	Cellulose Ether (kg)	W/B	PP Fiber (%)	BF Fiber (%)
	Cement	Coal Ash	Slaked Lime						
K0	0.75	0.2	0.05	1.1111	0.02	0.005	2.7	0	0
PP1	0.75	0.2	0.05	1.1111	0.02	0.005	2.7	0.3%	0
PP2	0.75	0.2	0.05	1.1111	0.02	0.005	2.7	0.6%	0
PP3	0.75	0.2	0.05	1.1111	0.02	0.005	2.7	0.9%	0
PP4	0.75	0.2	0.05	1.1111	0.02	0.005	2.7	1.2%	0
BF1	0.75	0.2	0.05	1.1111	0.02	0.005	2.7	0	0.3%
BF2	0.75	0.2	0.05	1.1111	0.02	0.005	2.7	0	0.6%
BF3	0.75	0.2	0.05	1.1111	0.02	0.005	2.7	0	0.9%
BF4	0.75	0.2	0.05	1.1111	0.02	0.005	2.7	0	1.2%

Note: Fiber mixing amount according to the percentage of the total mass of the cementitious material.

**Table 6 materials-17-03220-t006:** Compressive strength test program.

Compression Test (P)	X-Direction Force	Y-Direction Force	Z-Direction Force
Casting	-	-	-
Path a	P-X-a	P-Y-a	P-Z-a
Path b	P-X-b	P-Y-b	P-Z-b

**Table 7 materials-17-03220-t007:** Flexural Strength Test Program.

Flexural Test (K)	X-Direction Force	Y-Direction Force	Z-Direction Force
Casting	-	-	-
Path a	K-X-a	K-Y-a	K-Z-a
Path b	K-X-b	K-Y-b	K-Z-b

**Table 8 materials-17-03220-t008:** Thermal Conductivity Test Program.

Thermal Conductivity (D)	X-Direction	Y-Direction	Z-Direction
Casting	-	-	-
Path a	D-X-a	D-Y-a	D-Z-a
Path b	D-X-b	D-Y-b	D-Z-b

## Data Availability

The original contributions presented in the study are included in the article, further inquiries can be directed to the corresponding author. The data presented in this study are available in the article.

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
