# Peer review of "Study on the Effects of Printing Process and Reinforcement Materials on the Performance of 3D-Printed Glass Bead Insulation Mortar"

_materials, 2024, doi:10.3390/ma17133220_

Round 1

Reviewer 1 Report

Comments and Suggestions for Authors

The manuscript titled "Study on the effects of printing process and reinforcement materials on the performance of 3D printed glass bead insulation mortar" aimed to investigate how the printing process and reinforcement materials influence the performance of glass bead thermal insulation mortar in 3D printing. The study seeks to improve and enhance the performance of the mortar, providing experimental support for the application of 3D printing technology in the field of thermal insulation. The manuscript needs to be rewritten and reorganized in such a way that the information is clear to the reader and allows for review. Below are some observations:

·       The abstract needs to be completely rewritten. Present a sentence contextualizing the topic of the manuscript, state the main objective, the reason for conducting the experiments and characterizations, present the main experimental results, and indicate how these results contributed to the research field.

·       Change the verb tense of the text. For example, in line 87, change "is" to "was."

·       Read the "Instructions for Authors" and reorganize the titles of the "Materials and Methods" section. Also, renumber the table and figure captions according to the recommendations in the "Instructions for Authors."

·       Rewrite the text "Materials: P.O42.5 ordinary silicate cement produced by Zhucheng City Yangchun Cement Co., Ltd. is used, and the chemical composition is shown in Table 2-1; fly ash: Class I fly ash produced by Henan Hengyuan New Material Co. Chemical Technology Co., Ltd; re-dispersible rubber powder: Hongsheng Environmental Protection New Material Co." paying attention to punctuation according to English grammar.

·       The authors need to cite the model and manufacturer of the equipment used in the research.

·       Additionally, I recommend that the equipment and its respective descriptions be mentioned during the experiment description. Please check the articles already published in Material MDPI.

·       Photos of the equipment are not necessary in the manuscript. They should only be included if the equipment is self-made.

·       In the following text "The test is based on the printing performance of glass bead insulation mortar prepared by the group (water-to-cement ratio of 2.7, glue-to-sand ratio of 0.9, fly ash doping of 20%, slaked lime doping of 5%, and glue powder doping of 2%), and analyzes the effect of anisotropy on the 3D printing of glass bead insulation mortar through the compression test, flexural test, and thermal conductivity test of the printable glass bead insulation mortar. mechanical properties and thermal insulation properties," cite the scientific article where this information has already been published, otherwise, describe it.

·       It is not possible to understand the manuscript in its current format. Please read the "Instructions for Authors" section carefully and reorganize the manuscript. Primarily, include the topics "Materials and Methods" and "Results and Discussion."

Author Response

Thank you very much for reviewing my article, the questions you asked about my article helped me a lot, in the reply letter I answered your suggestions to me line by line and made changes, thank you again!

Reviewer 2 Report

Comments and Suggestions for Authors

The manuscript „Study on the effects of printing process and reinforcement materials on the performance of 3D printed glass bead insulation mortar “ deals with the printing process set the same printing parameters and studies the an isotropy of the two different printing paths,. The test is conducted on the specimen block according to the X, Y, Z three axes for marking, compared to the compressive and flexural strength are lower than that of the specimen block without 3D printing. Because it is difficult to realize the synchronous placement of steel reinforcement in the printing process, the original proportion based on the addition of reinforcing materials was used to study the effect of different fibers of different dosages on the performance of 3D printed glass beads thermal insulation mortar, to study and arrive at a thermal insulation mortar with good printability, excellent thermal insulation and high strength.

The study is sufficiently scientifically interesting and original in its technical content to justify publication. However, there are some errors that should be corrected prior to publication.

The study is topically appropriate for the Materials Journal.

There are no errors in fact, logic, or interpretation that require rectification. The paper is well-written, and the presentation is straightforward and concise. There is no need to delete, recast, or condense. Literature is adequate. There is a clear and straightforward link between experimental work, findings, and discussion.

Specific comments:

The title is informative, and it appropriately describes the theme of the study. Key words are adequate.

The abstract is somewhat informative. It does not fully describe the experimental work. Also, the gap in knowledge and the scientific novelty of the work should be highlighted.

The Introduction chapter provides a good preview of what is 3D printing and how it is applied in construction industry. This chapter also gives a preview of problems regarding printed reinforced materials. The work of other authors is presented and compared to the main idea of this study. The gap in knowledge and the scientific novelty of the work should be clearly outlined in the last paragraph.

Materials and methods are adequately presented. What methods (instrumentation) were used for characterization of the raw materials? Or are these data obtained from the manufacturer?

Mix design of the experimental mortar should be presented in a table.

Methodology of the printing process is elaborately described. The test is repeatable.

Results and discussion are adequate. The results are clearly presented in either diagrams or tables. All figures and tables are necessary. The connection between physic-mechanical characteristics and instrumental analyses is provided.

Table 3-3. “Different fiber dosage mortar mixing ratios” should be in Materials and methods section. Please consider slightly rearranging text.

Microstructural analysis of the obtained (printed) materials would be interesting. However, it can be an idea for further research.

Conclusions logically follow the results and discussion chapter. However, this section should be presented more as highlights than explanation.

References are up-to-date.

Comments on the Quality of English Language

Minor editing of English language required

Author Response

(The authors gave the same response as above.)

Round 2

Reviewer 1 Report

Comments and Suggestions for Authors

This is the second revision of the manuscript "Study on the effects of printing process and reinforcement materials on the performance of 3D printed glass bead insulation mortar." The authors have addressed many questions; however, there are still points to be clarified in the manuscript. See below:

·         The abstract contains two long sentences and needs to be rewritten. Suggested structure for the abstract:

o   First sentence: comment on general aspects of the manuscript's topic.

o   Second sentence: present the manuscript's objective.

o   Third sentence: present the main experiments conducted to achieve the manuscript's objective(s).

o   Fourth sentence: present the main results of the manuscript.

o   Fifth sentence: indicate how your results contribute to the global context of the manuscript's topic.

Please follow the Materials MDPI rules for the abstract, such as word count.

·         Reformulate (add more and change some) the keywords of the manuscript. The current ones do not adequately describe the presented topic.

·         Please correct the English grammar of the text between lines 103-106, especially the presentation of the tables.

·         Line 111, correct "Apparatuses."

·         Please see the "Instructions for Authors" section on the Materials MDPI website and check how to properly number the tables and figures.

·         What does "NO." mean in tables 2-4 and 2-5?

·         Line 135. What does "K0" mean?

·         The purpose of the "Materials and methods" section is to present the experimental parameters, experimental procedures, and equipment used in the research to ensure reproducibility. Therefore, any discussion that is not solely presenting these conditions is unnecessary. Hence, the text between lines 142 and 162 can be removed. The item "2.3 Setting of printer parameters" should be rewritten to provide only the necessary information to ensure the reproducibility of the work.

·         The item "3. Results" should be renamed to "3. Results and discussion."

·         Regarding the text "Compared with path a, path b significantly increased the compressive strength of the printed specimens in the X, Y, and Z directions, which may be attributed to the fact that the interlayer vertical printing reduces the gaps and weak surfaces of the specimens and makes the printed specimens denser inside." This is not clear in the figure; indeed, the results seem similar. Therefore, it is recommended that the authors show the uncertainties of the experimental data in the respective figure.

·         Show the uncertainties of the experimental data for all figures.

Author Response

Thank you very much for your review of my manuscript and very useful suggestions.

Reviewer 2 Report

Comments and Suggestions for Authors

The manuscript has been sufficiently improved to warrant publication in Materials.

Author Response

Thank you so much for reviewing my manuscript!

Round 3

Reviewer 1 Report

Comments and Suggestions for Authors

The authors did not answer all the questions of the reviewer.

The authors did not number the figures according to the Materials MPDI standard. According to the "Instructions for Authors" section, figures must be named “Figure 1”, “Figure 2”, “Figure 3”, etc.

The authors did not include the uncertainties of the experimental data in all figures in the manuscript.

Author Response

Thank you very much for reviewing my article and making useful suggestions, you were responded to point by point and highlighted in word. Thank you again.
